# The Concept of Intrauterine Programming and the Development of the Neonatal Microbiome in the Prevention of SARS-CoV-2 Infection

**DOI:** 10.3390/nu14091702

**Published:** 2022-04-20

**Authors:** Martina Grot, Karolina Krupa-Kotara, Agata Wypych-Ślusarska, Mateusz Grajek, Agnieszka Białek-Dratwa

**Affiliations:** 1Department of Epidemiology, Faculty of Health Sciences in Bytom, Medical University of Silesia in Katowice, 41-902 Bytom, Poland; s75443@365.sum.edu.pl (M.G.); awypych@sum.edu.pl (A.W.-Ś.); 2Department of Public Health, Department of Public Health Policy, Faculty of Health Sciences in Bytom, Medical University of Silesia in Katowice, 41-902 Bytom, Poland; mgrajek@sum.edu.pl; 3Department of Human Nutrition, Department of Dietetics, Faculty of Health Sciences in Bytom, Medical University of Silesia in Katowice, 41-808 Zabrze, Poland; abialek@sum.edu.pl

**Keywords:** microbiome, intrauterine programming, immunomodulatory factors, pregnant woman, newborn, SARS-CoV-2

## Abstract

The process of intrauterine programming is related to the quality of the microbiome formed in the fetus and the newborn. The implementation of probiotics, prebiotics, and psychobiotics shows immunomodulatory potential towards the organism, especially the microbiome of the pregnant woman and her child. Nutrigenomics, based on the observation of pregnant women and the developing fetus, makes it possible to estimate the biological effects of active dietary components on gene expression or silencing. Nutritional intervention for pregnant women should consider the nutritional status of the patient, biological markers, and the potential impact of dietary intervention on fetal physiology. The use of a holistic model of nutrition allows for appropriately targeted and effective dietary prophylaxis that can impact the physical and mental health of both the mother and the newborn. This model targets the regulation of the immune response of the pregnant woman and the newborn, considering the clinical state of the microbiota and the pathomechanism of the nervous system. Current scientific reports indicate the protective properties of immunobiotics (probiotics) about the reduction of the frequency of infections and the severity of the course of COVID-19 disease. The aim of this study was to test the hypothesis that intrauterine programming influences the development of the microbiome for the prevention of SARS-CoV-2 infection based on a review of research studies.

## 1. Introduction

The microbiome is the totality of genetic information of microorganisms located in the lumen of the gastrointestinal tract, as well as on the skin of the mother and the baby. The formation of the neonate’s gut microbiota through colonization of the microbial genome (microbiome) takes place in the internal (fetal) environment of the mother and during the method of natural childbirth [1]. On the other hand, intrauterine (fetal, metabolic) programming, which participates in the development of tissues and organ systems, begins from the moment of embryo and fetus formation. The concepts of microbiota and fetal metabolic programming share common features and a series of interactions as presented in Figure 1.

The relationship between these components is addressed by the scientific field of nutrigenomics. The diet and nutritional status of a pregnant woman, as well as environmental exposures, can affect the microbiome of the newborn. Intrauterine programming allows for the adjusting of the fetal intrauterine environment about metabolic and hormonal changes, and thus the maintenance of homeostasis. The hypothesis explaining the effect of microbiome programming is shown in Figure 2. Long-term adverse changes may contribute to the development of chronic disorders in the newborn’s body, and also, due to the progression of SARS-CoV-2 virus infection, predispose it to inactivate the immune system mechanism [3].

## 2. Review Methodology

This study aimed to investigate the hypothesis that intrauterine programming influences the development of the microbiome in terms of the prevention of SARS-CoV-2 infections. The review of scientific evidence was based on the available literature by entering sample phrases: intrauterine programming, microbiome, dysbiosis, SARS-CoV-2, COVID-19 (and various configurations and combinations) using a methodological tool in the form of the PubMed database. The literature search yielded 2385 records, from which 534 sources directly relevant to the topic of the paper were selected, and then those with the highest scientific value were selected according to bibliometric impact factors. The final literature review was based on seventy-three sources, representing the scientific output of the last two years (Figure 3).

## 3. Formation and Importance of the Neonatal Microbiome

The neonate’s immune system performs several defensive actions when colonization of probiotic microorganisms with higher metabolic potential predominates over pathogenic strains. Proven immunomodulatory effects are demonstrated by strains of the genus *Lactobacillus* spp., *Bifidobacterium* spp., *Bacteroides* spp., *Prevotellaceae*, *Firmicutes*, *Lachnospiraceae*, *Ruminococcacea*, and *Enterobacteriaceae* by stimulating the digestion of lactose disaccharide. The microbiome, due to the presence of a diverse genome with protective properties defined as strains targeting the digestive process, focuses on the conversion of complex substances (including dietary fiber and resistant starch) into simple components. As a result of biochemical transformations (fermentation) of other nutrients (complex saccharides), intestinal epithelial cells (colonocytes) via microbes (bacteria) receive energy resources in the form of short chain fatty acids (SFCA)-with particular emphasis on butyrate, propionate, and acetate [1,4]. As a result, short chain fatty acids stimulate the mechanism of absorption of such ions as Mg^2+^, Ca^2+^, and Fe^2+^, while microorganisms produce the following vitamins: vitamin K, thiamine (B_1_) riboflavin (B_2_), niacin (B_3_, PP), pyridoxine (B_6_), and cobalamin (B_12_) [1,5].

In intrauterine conditions, the first changes in the microbiome of the fetus occur until the period of birth defined as the moment of postnatal development, in which further development of the microbiota takes place. The mechanism of shaping the individual microbiome of each newborn is influenced by the clinical image of a pregnant woman, in particular the course of chronic health disorders and genetic predisposition, chronic antibiotic therapy, as well as an anthropometric parameter—body mass index (BMI) over 30 kg/m^2^-obesity, as well as the type of chidbirth, skin-to-skin contact and the way your baby is fed [5]. The mentioned characteristics direct the pathogenic process of microbiome formation in the fetus and child in the postnatal phase [1]. Moreover, the relationship between the antigenic components of the mother’s microbiota in utero and the potential for an immune response on the fetal side, as well as the influence of the individual state of the pregnant woman’s microbiota on the state of the microbiome of the newborn child seems to be of interest [5,6]. The quantitative and qualitative characteristics of the newborn’s microbiota depend on internal factors (uterine environment including the state of the maternal microbiota, the woman’s feeding pattern, the pregnant woman’s age, fetal/metabolic programming) and external factors such as the environment, method of delivery, home conditions and the way the infant is fed. In the light of scientific reports, we can conclude that the microbiota of the newborn in the first week of life is characterized by the following types of probiotic bacteria: *Enterococcacae*, *Clostridiaceae*, *Lactobacillaceae*, *Bifidobacteriaceae*, and *Streptococcaceae* [5,6,7,8] (Figure 4).

Factors predisposing to stimulation of the mechanism of intrauterine programming constitute response of adaptation of the environment inside the organism and activate a series of reactions leading to intrauterine balance. Because of the risk of pathological changes, pregnancy lasting more than 42 weeks may adversely affect the health of the newborn, even until adolescence. The factors affecting the mechanism of fetal programming is shown in Figure 5 [3,9,10,11,12].

## 4. Nutritional Status and Diet of the Pregnant Woman Protectively Affect the Gut Microbiota of the Newborn

Diet therapy is essential in the prevention and minimization of the development of chronic diseases in children. Due to its immunomodulatory properties, food of a pregnant woman is a form of a synbiotic with the child’s intestinal microbiota. Due to this fact, the nutritional status of a woman planning pregnancy and, above all, during pregnancy, is an important and priority element in the prevention of physical and mental health. It should be assessed in terms of appropriate laboratory and immunological markers to diagnose malnutrition, as well as overnutrition (overweight and obesity) in the form of metabolic profile parameters, as well as microbiological tests and microflora of the intestines to implement personalized probiotic therapy allowing for the stimulation of the microbiotic programming process. A secondary issue is the balanced nutrition of the mother-to-be and the pregnant woman during the ongoing pregnancy, namely the presence of components with the high bioactive potential of compounds present in nutritional products. The bioactive substances in the nutritional model of the pregnant woman that have a health-promoting effect on the whole clinical picture of the newborn, especially in the first 33 months after birth, consist of folic acid, iodine, iron, magnesium, calcium, vitamin B_12_, flavonoids, polyphenols, antioxidants in the form of vitamin E, C, retinol (especially following dietary recommendations-excess has a toxic effect on the fetus) lycopene, omega-3, and omega-6 acids in predominant amounts of saturated fats. Based on scientific reports developed by specialists in clinical nutrition, a model of nutrition referred to as the Mediterranean diet-DASH (Dietary Approaches to Stop Hypertension) based on vegetable oils (flaxseed oil, avocado, nigella, cashew nuts, rapeseed) vegetables with no or short heat treatment, whole grains with low glycemic index and glycemic load and whole-grain products rich in soluble (colonizing probiotic bacteria) and insoluble (minimizing the risk of colonic diverticulosis) fiber are immunomodulatory factors. Furthermore, the implementation of prebiotics, the source of which are onions, garlic, tomatoes, chicory, asparagus, artichokes, peanuts, and bananas, show an immunoprotective effect on the microbiome of the pregnant woman as well as the fetus, and subsequently the newborn [13,14,15,16,17,18,19,20,21,22,23].

## 5. The Importance of the Microbiome in Terms of the Prevention of SARS-CoV-2 Virus Infection

Exposure to external factors, especially in the form of pathogenic microorganisms from the group of bacteria (eg., *Clostridioides difficile*) and viruses (eg., SARS-CoV-2) predisposes the development of infections in the fetus by disturbing the gastrointestinal microbiome. Epidemiological data based on nationwide case-control studies among patients with a history of gastrointestinal infections indicate increased development of microbial dysbiosis causing inflammation within the intestinal cells [24,25,26]. During pregnancy at the fetal-placental level, environmental pathogens contribute to a state of malnutrition, increased levels of dysbiosis, and their translocation in the enterocyte mucosa, while contributing to the formation of a dysbiotic microflora in the fetus. Expanding studies using metabolomics and transcriptomics techniques will allow for environmental metabolic analysis during the fetal and postnatal period of the infant. Gastrointestinal inflammation occurs in most children before five years of age, contributing at a rate of 1 million annual deaths in the pediatric population to worldwide. According to microbiological analyses, the mixed type of viral-bacterial infections was associated with individual microbiota status characterized, among others, by the abundance of strains of the genus *Bifidobacteriaceae* [27,28,29]. The dysbiotic state of the intestinal microbiota resulting from the extended diagnosis influences the increased frequency of the initiation of the infectious process in the child’s body, consequently disrupting the intrinsic environmental homeostasis of individual cells and tissues. This process is complex because the functional scope of the intestine focuses both on the defense barrier against pathogens and is the site of attachment of microorganisms and pathogen associated molecular patterns (PAMPs). The pathomechanism of pathways leading to dysbiosis targets immune cell activation, epithelial barrier destruction, intestinal villi atrophy, and crypt hypertrophy. Pathophysiological changes contribute to the activation of proinflammatory cytokines and the formation of the so-called cytokine storm, a proliferation of T lymphocytes in an impaired manner, causing systemic inflammation, secretion of systemic endotoxin, a reduced nutrient absorption surface within the intestine, and increased intestinal permeability with the movement of PAMPs in the form of lipopolysaccharides [30,31]. The immune system under the influence of weakness inactivates the range of immunomodulatory functions of the microbiome, which mediates the regulation of the processes of metabolism, transport, and the absorption of proteins, fats, and carbohydrates, monitoring inflammation. Pathological changes in the quantitative (number and proportion of microbial strains) and qualitative (species diversity of strains) structure of the intestinal flora exacerbate dysfunction in the integrity of the intestinal mucosa, increasing the progression of viral infections in particular. A meta-analysis of clinical studies shows that the state of the microbiota during COVID-19 disease undergoes significant destructive changes as a result of the immune response. They are noticeable in the image of biochemical parameters in the form of an increased level of: neutrophils, interferon, TNF-α (Tumor Necrosis Factor), IL-1 (interleukin 1), IL-6 (interleukin 6), IL-12 (interleukin 12), IL-18 (interleukin 18), adiponectin, and D-dimers. On the other hand, the neutral amino acids L-DOPA (levodopa), tryptamine, β-PEA (B-phenylethylamine) as well as Th 2 hyperreactivity and T reg deficiency were down-regulated. These changes also significantly contributed to the elevation of other inflammatory markers in the form of HOMA (Homeostatic Model Assesment), AST (aspartate aminotransferase), ALT (alanine aminotransferase), and IGF-1 (insulin-like growth factor 1) index. Subsequently, the process of virus replication caused the multiplication of *Enterobacteriaceae* and *Enterococcus* strains, the ratio of which above three influenced the degree of SARS-CoV-2 viral infection, then reducing the number of probiotic strains of *Bifidobacerium*, *Lactobacillus*, *Faecalibacterium*, and *Roseburia with the development of bacteria Ruminococcus gnavus*, *Bacteroides vulgatus*, *Tectiviridae*, *Granulicatella* and *Rothia mucilaginosa* (oral cavity and intestines). Subsequently, lesions of the pathogenic opportunistic strains *Clostridium ramosum*, *Coprobacillus*, *Clostridium hathewayi*, *innocuum*, *Streptococcus*, *Veillonella*, *Actinomyces naeslundii*, and *Erysipelatoclostridium* were visualized, while a diagnosis of *Faecalibacterium prausnitzii* and *Clostridium leptum* showed significantly increased neutrophil levels and the *Eubacterium rectale* strain increased IL-6 levels. Stool diagnostics performed among patients with COVID-19 diagnosis indicate that 50% of patients have virus replication within the gastrointestinal tract, indicating the presence of calprotectin, IgA (immunoglobulin A), lipocalin, IL-8, IL-18, neutrophils in the stool, and the frequency of gastrointestinal symptoms in the form of diarrhea as an indicator of inflammation, as well as an abnormal microbiota before the virulence process of SARS-CoV-2. The presence in the microflora of strains of the genus *Bacteroidetes* and *Firmicutes* allows for the expression of the angiotensin 2 (ACE2) enzyme receptor, which is involved in the virulence of SARS-CoV-2 into body cells and activated by small intestinal enterocytes leading to interaction with ACE2 as a result of the secretion of antimicrobial peptides and consequently inducing a pro-inflammatory bidirectional immune response at the level of the gut-lung axis (GLA). The intestine is an active site of viral replication, and viral entry also occurs through the presence of the enzyme trans-membrane serine protease 2 and 4-TMPRSS2 and TMPRSS4 in erythrocytes, which together with ACE2 enhances microbial proliferation, and induces autophagy leading to the symptom of diarrhea due to dysbiosis. Moreover, SARS-CoV-2 virus proliferation in the intestine is associated with the modulation of heterogeneous bacterial species-increasing the likelihood of a reduced antiviral immune response and exacerbating the progression of systemic inflammation. In the light of scientific reports, the presence of an intestinal vortex in the form of 20–21% of viral DNA from the family Myoviridae, Siphoviridae in the course of COVID-19 as a result of replication of the SARS-CoV-2 virus has been proven. The course of direct pathological changes focuses on the induction of apoptosis of infected enterocytes or indirectly through quantitative and qualitative changes in the microbiota, which induce inflammation in the brush border of the intestine, deactivating the metabolic pathway of anti-inflammatory reactions [31,32,33,34,35,36,37,38,39,40,41,42,43]. A three- to six-month follow-up period after COVID-19 infection in adults indicates low levels of restoration of intestinal eubiosis from either acute or chronic dysbiosis, suggesting a longer-term therapeutic effect of the microbiome status.

Due to the asymptomatic course of SARS-CoV-2 virus among children, we should analyze the condition of their microbiota because of the strong predisposition to the development of autoimmune, metabolic diseases after viral infection and the disruption of enterocyte maturation. The mechanism of autoimmunity is a marker that determines the degree of disease progression and severity and clinical improvement through the balance of the GLA axis after COVID-19 disease. The development of intestinal dysbiosis and changes in microbiota homeostasis contribute to the reduction of the immune system response, predisposing to an increase in risk factors related to SARS-CoV-2 virus replication, and thus to the induction of the secondary inflammatory infection mechanism. The secretion of metabolites of pathogenic microorganisms with the previous dysbiosis because of viral infection shows a reciprocal effect at the bidirectional level towards the microbiome of the brain-gut-lung axis [43,44,45,46,47,48,49,50,51,52,53]. The degree of permeability of pathogens across the intestinal barrier is crucial on the pathomechanism of as seeping gut syndrome and viral infections. The transmission process depends on approximately 31 inflammatory markers, the diversity of the pathogenic versus the probiotic microbiome, and the enzymatic cascade. As a result of changes leading to leaky gut, the integrity of the barrier between intestinal cells is lost, mucus secretion decreases, the mucosal layer that has a protective function, and partial inactivation in the epithelial layer of proteases, immunoglobulin A, and antimicrobial peptides. These mechanisms lead to increased penetration of pathobiotics and pathogens by the destructive breach of the immuno-intestinal barrier to the surface of enterocytes due to a broad spectrum of pathogenic microorganisms. Next, the process of SARS-CoV-2 virus penetration through the leaky mucosal layer and fewer connections between intestinal cells is directed at the possibility of binding the virus to the angiotensin-converting enzyme 2 (ACE2) present in enterocytes, intensifying the state of “leaky gut” and activation of the virus S protein. Consequently, it penetrates through the gastrointestinal tract into the bloodstream and other organs [54,55]. In the prevention of primary and secondary viral infections it is important to implement probiotic therapy as a health-promoting element of pharmacological and non-pharmacological therapy. Live protective strains of specific groups of microorganisms from the bacterial family used in the form of probiotics show, among other things, a two-stage action directed at:Therapeutic effect, neutralizing quantitative and qualitative inflammatory changes due to the progression of intestinal dysbiosis, saturated bowel syndrome also reducing the duration of viral and bacterial infections.Preventive potential against the development of a leaky intestinal barrier and complete multiplication of a qualitatively and quantitatively pathogenic microbiome, especially in children.

The pathophysiological mechanism focuses on the immunomodulatory properties of the body’s microbiota, in which a specific probiotic preparation is applied, initiating the pathway of activation of morphotic elements in the form of a high concentration of T, suppressor and helper type lymphocytes (with a decrease in B lymphocytes), and immunoglobulins of A, G, M class in saliva, interferon alpha, interleukin 10—inhibition of the pro-inflammatory cytokine response. This is followed by the process of activation of NK (natural killer) cells, T cells, and Paneth cells, stimulating the secretion of antimicrobial peptides on the example of cathelicidin (peptide LL-37 (antimicrobia peptid) in the lower segment of intestinal crypts and the process of differentiation of Th17 cells and activation of the TLR4 (toll-like receptor 4) receptor in the gastrointestinal tract. Scientific reports indicate the strong antiviral potential of probiotics against SARS-CoV-2 virus infection with different strains of the pathogen classified as coronavirus [56,57]. In probiotic preparations, the presence of strains with proven antimicrobial and antiviral effects shows normalization of Ph, synthesis of B vitamins, phylloquinone (K), and intestinal calcium absorption bioavailability, among others. The types of immunomodulatory strains is shown in Figure 5. Their action is directed at strengthening the immune system or altering the microflora within the colon or other segments of the intestine [57,58,59,60].

Current evidence suggests that immunobiotics (probiotics) have protective properties aimed at reducing the frequency of infections and the severity of COVID-19 disease (Figure 6). However, it is necessary to conduct multiple randomized double-blind clinical trials in a large study group and their subsequent meta-analyses to determine the amount of probiotic implemented with its dosing period along with special attention to the number identifying a specific collection of strain culture with widely proven, individualized and strongly confirmed therapeutic potential [61,62,63,64].

The Mediterranean model of dietary therapy, which includes the use of dietary fiber in the form of soluble fruits and vegetables, inulin, resistant starch, prebiotics (kale, garlic, onions, oats, asparagus, banana), and monounsaturated fatty acids (MUFA) (olive oil, soybean oil-reduction of *Bacteroides* spp., *Proteobacteria*, *Desulfovibrionaceae*, *Escherichia* and *Streptococcus*), bioactive components in the form of polyphenolic compounds derived from green tea due to the presence of epigallocatechin 3-gallate (EGCG), synbiotics enhancing the nutritional therapy strategy with antioxidant and antiviral (replication inactivation), antibacterial, anti-inflammatory potential [65,66]. Probiotic preparations containing strains classified as psychobiotics show the protective activity of the vagus nerve. When implemented for at least three months, they contribute to the modification of the expression of the GABA (Gamma-aminobutyric acid) receptor, which predisposes people to an anxiety-depressive pathology. On the other hand, the allostatic load, which disturbs the intra-body balance by adapting to stress factors, contributes to the development of metabolic and cardiological disorders.

According to the literature, the safest psychobiotics (with proven clinical effect) for a pregnant woman’s health and, consequently, for her child, are *Bifidobacteruim* longum Rosell-175 and *Lactobacillus* helveticus Rosell-52. In addition, the Natural and Non-prescription Health Products Directorate in Canada guidelines assume the use of combined forms of two psychobiotics, aimed at reducing anxiety symptoms, maintaining emotional stability and minimizing symptoms in the digestive system caused by stress factors. The supplementation of psychobiotics with their probiotic potential in the form of fermented products (soybeans, rice bran, milk, black soy milk, yogurt) during pregnancy, may contribute to the reduction of oxidative stress at the level of the gut-brain axis, and consequently influence the proper condition of the fetal intestinal microflora and its psychological well-being [21,22,23].

In the light of scientific research, bioactive nutrients such as active form of vitamin D (1,25(OH)_2_ D at the level ≥30 ng/mL), vitamins K, E, and A, together with beta-carotene, cobalamin, pantothenic acid, folic acid, phytotherapeutics containing mainly organic acids, alkaloids, flavonoids, phenylpropanoids, and glycosides (e.g., ginseng, curcumin, glycosides, etc.), have a confirmed immunostimulating effect. Phytotherapeutics mainly contain organic acids, alkaloids, flavonoids, phenylpropanoids, glycosides, e.g., ginseng, curcuma (curcumin-causes the addition of viral protease receptors), glycyrrhizic acid, resveratrol, bromelain, phenolic compounds (whole-grain products) essential amino acids (increase in the ratio of vegetable to animal protein), iron, copper, zinc, selenium, omega-3 acids: ALA—The precursor to eicosapentaenoic acid (EPA) and docosahexaenoic acid (DHA), prebiotics galactooligosaccharides and fructooligosaccharides, fermented products-vegetables, kimchi, yogurt, kefir, miso, tempeh, kombucha and fermented drink apple cider, beetric acid. Scientific evidence emphasizes their preventive effect to a moderate degree on the induction of the course of SARS-CoV-2 virus infection, but they activate the immune system for antioxidant and probiotic effects (higher *Bacteroidetes*/*Firmicutes* strain ratio) dependent on the patient’s previous nutritional status and adequate immuno-neuro-hormonal interaction on the brain-gut-lung axis [67,68,69,70]. Among the alternative methods in extreme clinical situations caused by chronic intestinal dysbiosis to seal the intestinal mucosal barrier, the transplantation of “microbiologically safe” intestinal/fecal microflora subjected to a washing process before transplantation is applicable. Clinical studies confirm the therapeutic relevance of transplantation towards the preservation of eubiosis in the microbiome and the reduction of the bacterial and viral-mediated inflammatory process (SARS-CoV-2), as well as the enhanced immune response after COVID-19 vaccination (Table 1) [31,55,56,58,60,63,64,67,70,71,72].

The study explores the current and often underestimated topic of gut microbiota shaping and intrauterine programming. This is especially important during the COVID-19 pandemic. The performed synthesis emphasizes the role of the intestinal microbiota in shaping the immunity of adults, newborns and the developing fetus. Unfortunately, the research hypothesis put forward, which was aimed at answering the question of whether intrauterine programming influences the development of the microbiome in the field of SARS-CoV-2 infection prevention, cannot be unequivocally accepted. On the other hand, the analyzed scientific data do not justify rejecting the null hypothesis in favor of the alternative hypothesis, which is that intrauterine programming does not affect the development of the microbiome in terms of preventing SARS-CoV-2 infections. Most of the analyzed publications concerned studies conducted on a group of adults. However, they show the importance of the microbiome in COVID-19 disease as well as in shaping vaccine immunity. The relationship between microbiome formation and SARS-CoV-2 infection is a completely new issue. To date, no study of the gut microbiome of young children infected with COVID-19 has been conducted. Thus, the indicated limitation of this publication, mainly related to the analysis of data available for adults, seems justified and does not affect the quality of the collected scientific evidence. Moreover, using inference by analogy, we can speculate that the newborn’s microbiome will play a preventive role against various diseases that are not only autoimmune in nature, but are also infectious. Furthermore, knowledge about microbiota programming should be widely disseminated among groups of women, especially pregnant women, and not only available and discussed in scientific circles. Such review publications may become a tool of health education, which is also often a method of primary prevention.

## 6. Conclusions

The concept of intrauterine programming influences the state of the fetal gut microflora in a particular way by regulating the gut-brain-lung axis. In connection with the above-mentioned response to the hypothesis, such a prenatal nutrition method should be considered in the future, as this will allow for an immunomodulatory effect on the child’s microbiome. It is worth extending the diagnosis to include prenatal tests and a detailed medical interview of both parents as a preventive measure. Supplementation during fetal development, natural childbirth, as well as the method of breastfeeding and the lack of recognition of metabolic diseases (e.g., obesity, hybrid diabetes, various endocrinopathies, MTHFR polymphrism) with a balanced nutritional regimen in adolescence predestines the proper functioning of the microbiome and the child’s immune system [73].

## Figures and Tables

**Figure 1 nutrients-14-01702-f001:**
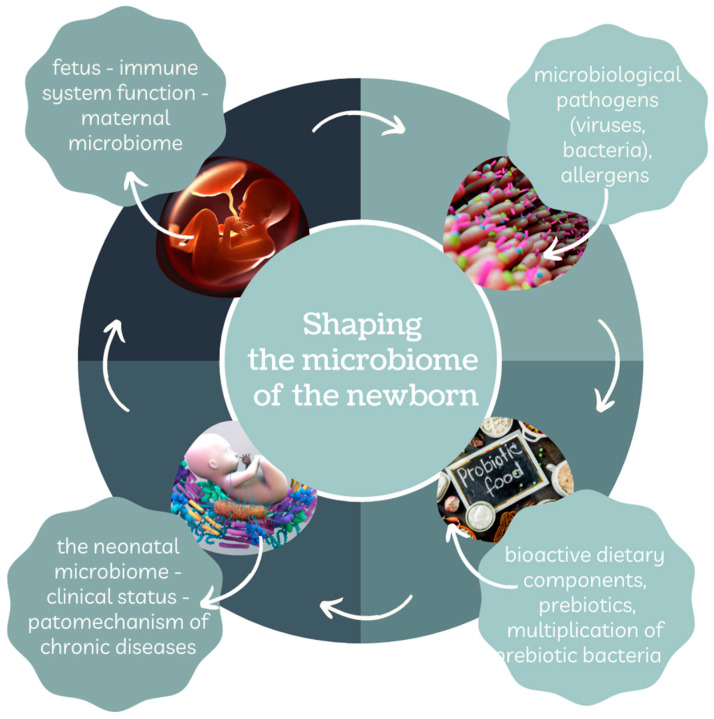
Shaping the newborn’s microbiome. Own elaboration based on [2].

**Figure 2 nutrients-14-01702-f002:**
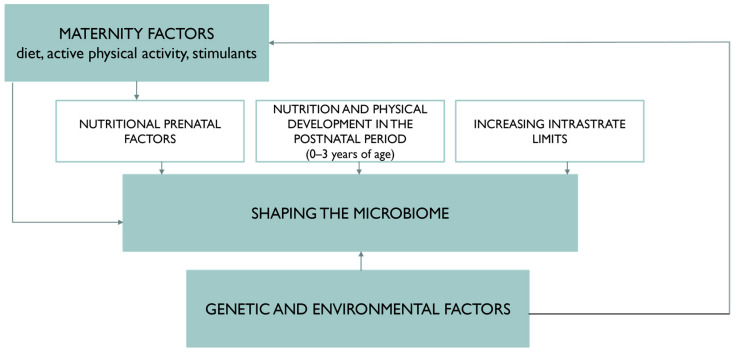
Hypothesis explaining the effect of microbiome programming.

**Figure 3 nutrients-14-01702-f003:**
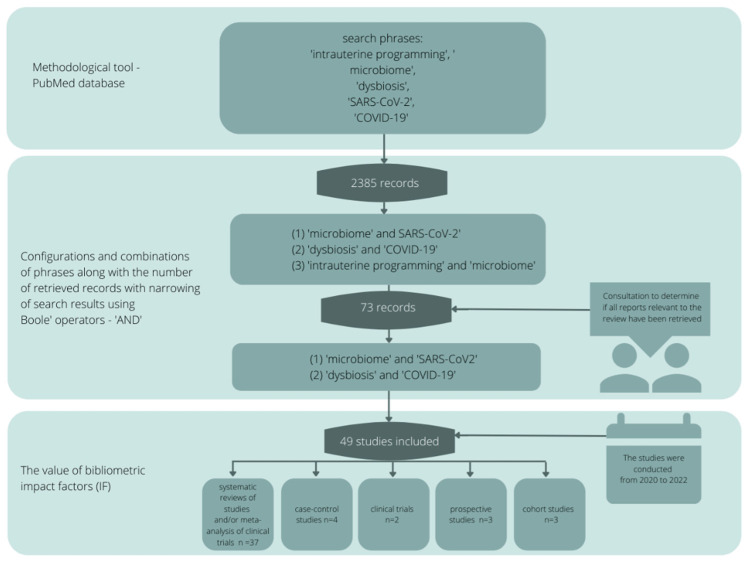
Publication search algorithm.

**Figure 4 nutrients-14-01702-f004:**
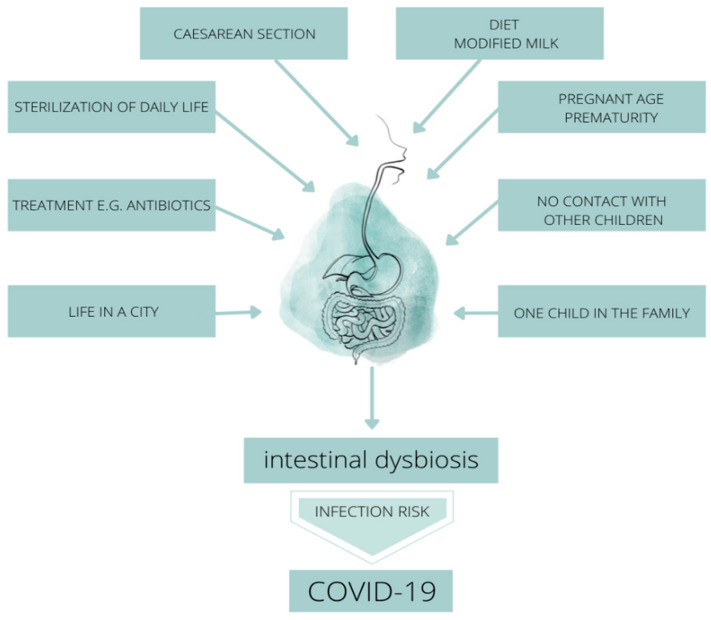
Factors influencing the development of intestinal dysbiosis in the period of shaping the intestinal microbiome in terms of the risk of SARS-CoV-2 infection.

**Figure 5 nutrients-14-01702-f005:**
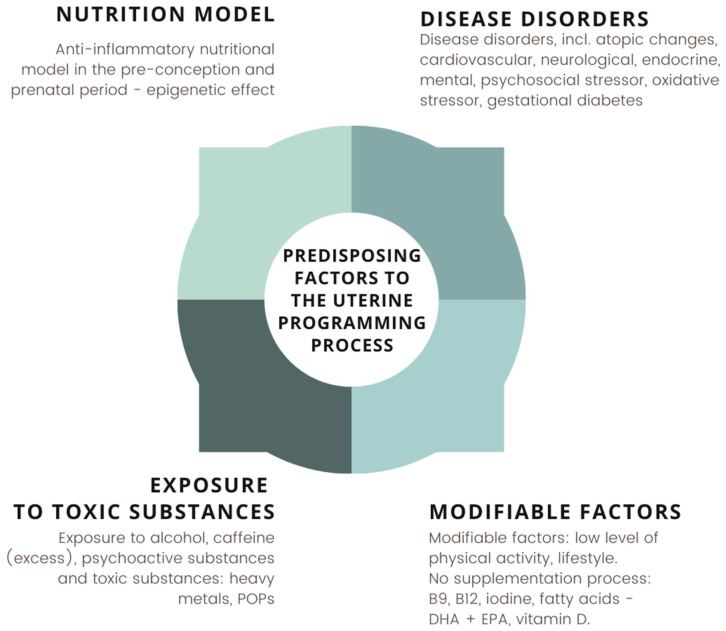
Factors predisposing to the uterine programming process in the critical phase of fetal development. Own elaboration based on [3,9,10,11,12].

**Figure 6 nutrients-14-01702-f006:**
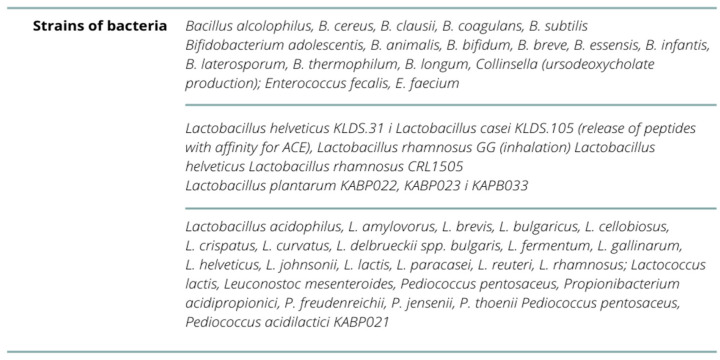
Strain cultures with widely proven, individualized and strongly confirmed therapeutic potential.

**Table 1 nutrients-14-01702-t001:** Impact of probiotic strains on reducing the risk of SARS-CoV-2 virus infection.

Impact on Reducing the Risk of Infection-Strain		
Authors	Increased (Probiotic Potential)	Reduced (Pathogenic Potential)
Li F. et al. (2020) [31]	*Firmicutes*, *Romboutsia*, *Faecalibacterium*, *Fusicatenibacter Eubacterium hallii*, *Faecalibacterium prausnitzii*	*Bacteroidetes*, *Streptococcus*, *Rothia*, *Veillonella*, *Erysipelatoclostridium*, *Actinomyces*, *Clostridium ramosum*, *Coprobacillus* and *Clostridium hathewayi*
Hu J. et al. (2021) [55]	*Faecalibacterium prausnitzii*, *Lachnospiraceae*, *Eubacterium rectale*, *Ruminococcus obeum* and *Dorea formicigenerans*, *Bacillus*, *Lactobacilli*, *Bifidobacteria*, *Lactococcus lactis*	*Clostridium hathewayi*, *Actinomyces viscous*, *Bacteroides nordii*, *Coprobacillus*, *Clostridium ramosum*
Olaimat AN. et al. (2020) [56]	*Lactobacillus acidophilus*, *L. amylovorus*, *L. brevis*, *L.bulgaricus*, *L. casei*, *L. cellobiosus*, *L. crispatus*, *L. curvatus*, *L.delbrueckii* spp. *bulgaris*, *L. fermentum*, *L. gallinarum*, *L.helveticus*, *L. johnsonii*, *L. lactis*, *L. paracasei*, *L. plantarum*, *L.reuteri*, *L. rhamnosus; Streptococcus thermophilus*, *Lactococcus lactis*, *Leuconostoc mesenteroides*, *Pediococcus pentosaceus*, *P. acidilactici*, *Bifidobacterium adolescentis*, *B.animalis*, *B. bifidum*, *B. breve*, *B. essensis*, *B. infantis*, *B.laterosporum*, *B. thermophilum*, *B. longum*, *Propionibacterium acidipropionici*, *P. freudenreichii*, *P.jensenii*, *P. thoenii*, *Enterococcus fecalis*, *E. faecium*, *B. alcolophilus*, *B.cereus*, *B. clausii*, *B. coagulans*, *B. subtilis*, *Escherichia coli*, *Sporolactobac*, *L. gasseri*, *L. delbrueckii* ssp. *yeast: Saccharomyces boulardii* and *yeast S. cerevisiae B. breve*, *L.pentosus*, *L. casei*, *L. plantarum*, *L. rhamnosus*, *L. delbrueckii* ssp. *bulgaricus*, *L. gasseri*, *L. reuteri*, *L. lactis i L. brevis*—given intranasally or orally	-
Shahbazi R. et al. (2020) [58]	*Lactobacillus*, *Bifidobacterium*, *Faecalibacterium prausnitzii*, *Lactobacillus helveticus*, *Lactobacillus casei*, *Lactobacillus acidophilus*, *Lactobacillus reuteni*, *Bifidobacterium bifidum and Streptococcus thermophilus*, *Candida kefyr*, *Bifidobacterium*, *Prevotella* and *Lactobacillus*, *Bifidobacterium longum subsp. infantis E4* and *Bifidobacterium breve M2CF22M7*, *Lactobacillus mucosae NK41*, *Bifidobacterium longum NK46*, *Lactobacillus reuteri NK33* and *Bifidobacterium adolescentis NK98*	*L. rhamnosus GG*, *Lactobacillus delbrueckii ssp. bulgaricus OLL1073R-1*, *Anaeroplasma*, *Rikenellaceae* and *Clostridium*, *C. butyricum*, *Lactobacillus casei DG*
Gutiérrez-Castrellón, P. et al. (2022) [60]	*Lactiplantibacillus plantarum KABP022*, *KABP023* and *KAPB033* and *strain Pediococcus acidilactici KABP021*	-
Ailioaie, LM. et al. (2021) [63]	*Lactobacillus*, *Bifidobacterium*, *Streptococcus*, *Pediococcus*, *Leuconostoc*, *Bacillus* and *Escherichia coli*, *Lactobacillus paracasei 28.4*, *L. reuteri-CFS*, *Lactobacillus casei CRL 431* and *Bacillus coagulans GBI-30*, *Lactococcus*, *L. acidophilus*, *Streptococcus thermophilus*	*Clostridioides difficile*, *Lactobacillus rhamnosus GG (LGG)* and *Bifidobacterium animalis subsp. lactis BB-12*, *Shigella*, *Salmonella*, *E. coli*, *Yersinia enterocolitica*, *Campylobacter jejuni*, *C. auris*, *Clostridium butyricum*, *Leuconostoc cremoris*, *Faecalibacterium prausnitzii*, *Eubacterium rectale*, *Bifidobacterium*
Shinde T. et al. (2020) [64]	*L. rhamnosus. B. lactis HN019*, *Bacillus coagulans BC30 PB*, *L.acidophilus DDS-1*, *Anaerostipes hadrus*	*B. infantis R0033*, *B. bifidum R0071* and *L.helveticus*—poorly proven beneficial effects
Jabczyk M. et al. (2021) [67]	*Roseburia*, *Lachnospira*, *Bificobacteria i Collinsella*, *Actinobacteria*, *Faecalibacterium prausnitzi*, *Bifidobacterium bifidum*, *Eubacterium ventriosum*, *Lachnospiraceae*, *Lactobacillus*, *Akkermansia*, *Firmicutes/Bacteroidetes*, *Lactobacillus rhamnosus GG*, *Bacillus subtilis*, *Enterococcus faecalis*, *Lactobacillus plantarum*, *Lactobacillus reuteri*, *Lactococcus lactis*, *Bifidobacterium infantis*, *Bifidobacterium animalis*	*Proteobacteria*, *Akkermansia muciniphila*, *Bacteroides dorei*, *Bacteroides nordii*, *Clostridium hathewayi* and *Actinomyces viscosus*, *Staphylococcus*, *Escherichia*, *Streptococcus*, *Lactobacillus acidophilus*, *Bacillus clausii*, *Fusobacterium*
Daoust L. et al. (2021) [70]	*Aecalibacterium prausnitzii*, *Lactobacillus rhamnosus GG kombinacje: Bacillus subtilis* and *Enterococcus faecalis*	*Listeria monocytogenes*

## Data Availability

Not applicable.

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
