# Peer review of "The Concept of Intrauterine Programming and the Development of the Neonatal Microbiome in the Prevention of SARS-CoV-2 Infection"

_nutrients, 2022, doi:10.3390/nu14091702_

Round 1
Reviewer 1 Report
This review is about the intrauterine programming in the prevention of SARS-CoV-2 focusing on nutritional intervention for pregnant women. Authors concluded that the use of probiotics could reduce the number of infections and the severity of the course of COVID-19.
There are some details that must be addressed:
-The point 5 “Psychobiotics and the gut-brain axis in a woman with an ongoing pregnancy about fetal development” should be eliminated and the content must be moved to next point in line 294. Regarding point 5, the reference 22 should be eliminated because is not correspond to the psychobiotics in the review context.
Author Response
Dear Reviewer,
Thank you for giving us the opportunity to make necessary adjustments. All suggested changes have been applied. We took a closer look at the text of the article with a native English-speaking colleague. We apologize for an earlier imperfect version.
This review is about the intrauterine programming in the prevention of SARS-CoV-2 focusing on nutritional intervention for pregnant women. Authors concluded that the use of probiotics could reduce the number of infections and the severity of the course of COVID-19.
There are some details that must be addressed:
- The point 5 “Psychobiotics and the gut-brain axis in a woman with an ongoing pregnancy about fetal development” should be eliminated and the content must be moved to next point in line 294.
Response: The content of item 5 has been moved to the line number 296 (the content of item 5 is located between lines 296-312).
- Regarding point 5, the reference 22 should be eliminated because is not correspond to the psychobiotics in the review context.
Response: The original reference has been replaced by a new, matching reference (493).
We hope that the changes made will allow you to accept this article for publication.
Best regards, Karolina Krupa-Kotara
Reviewer 2 Report
I read the manuscript "The concept of intrauterine programming and the development of the neonatal microbiome in preventing SARS-CoV-2 infection" with great interest, and I believe we need all we can learn from Sars-Cov-2 infection, especially in the pediatric population. But , I found the manuscript out of focus and rather construed.
- The methodology is not transparent and well explained
- The results are not systematically structured and are mostly general assumptions rather than focusing on Sars-Cov-2 infection in detail
- The conclusion is out of line with the hypothesis
Therefore I would recommend authors present their methodology in detail transparently, give the exact keywords/combinations and databases for the research, inclusion-exclusion criteria for the reviewed publications. A diagram would also be helpful.
I would also recommend summarizing the results from the included prospective study publications in a table.
I understand that Sars-Cov-2 infection is new, and the quality of the published data may be questionable. But that is exactly the reason we need reliable data included in those kinds of reviews not to mislead.
I thank the authors for this well-written manuscript; however, I kindly believe it need major revisions and the authors need point out all the issues and weaknesses of their work.
Author Response
Dear Reviewer
Thank you for giving us the opportunity to make necessary adjustments. All suggested changes have been applied. We took a closer look at the text of the article with a native English-speaking colleague. We apologize for an earlier imperfect version.
I read the manuscript "The concept of intrauterine programming and the development of the neonatal microbiome in preventing SARS-CoV-2 infection" with great interest, and I believe we need all we can learn from Sars-Cov-2 infection, especially in the pediatric population. But, I found the manuscript out of focus and rather construed.
- The methodology is not transparent and well explained.
Response: The methodology is explained in Figure 3 in the review methodology section (370-369).
- The results are not systematically structured and are mostly general assumptions rather than focusing on Sars-Cov-2 infection in detail.
Response: We have tried to adjust and systematize the results, as per your suggestion. We hope that You find the attached table 1 (366-369) hepful regarding result receival.
- The conclusion is out of line with the hypothesis.
Therefore I would recommend authors present their methodology in detail transparently, give the exact keywords/combinations and databases for the research, inclusion-exclusion criteria for the reviewed publications. A diagram would also be helpful.
Response: The methodology is presented in the form of a diagram including the exact keywords and their combinations along with the inclusion of specific types of studies (73-77). The answer to this hypothesis is presented in the “Conclusions” chapter (415-437). Methodology was corrected. Additionally, we implemented the suggested diagram, that includes the method of selecting publications for research as well as to help with interpretation.
- I would also recommend summarizing the results from the included prospective study publications in a table.
Response: A summary of studies on the effect of specific bacterial strains is given in Table 1 (366)
- I understand that Sars-Cov-2 infection is new, and the quality of the published data may be questionable. But that is exactly the reason we need reliable data included in those kinds of reviews not to mislead.
Response: Thank You for paying attention to this. We hope, that the change we implemented earlier will contribute to the improvement of quality of the study and match your expectations.
- I thank the authors for this well-written manuscript; however, I kindly believe it need major revisions and the authors need point out all the issues and weaknesses of their work.
Response: Thank you for the suggestion , we have tried to the best of our ability to show the limitations and strong points of this article (370-414).
We hope that the changes made will allow you to accept the article for publication.
Best regards, Karolina Krupa-Kotara